# PROBLEM-DEPENDENT QUANTUM CIRCUIT DESIGN BASED ON ENTROPY MATCHING

## ABSTRACT

Variational quantum machine learning (QML) have shown great promise for harnessing quantum advantage in machine learning tasks. However, architecture design of quantum circuits employed in these QML algorithms has been poorly explored for practical problems. Specifically, quantum circuits should have sufficient expressibility for modeling complex functions considering the inherent structures of real-world data. Naively increasing the circuit depth could enhance the expressibility of quantum circuits, which also induce the barren plateau problem as a by-product. In this work, we develop an architecture design framework to solve this problem. We use a simple yet effective metric of quantum entanglement, i.e. the linear entropy, to guide the circuit design from the perspective of the input data. First, we quantify the entanglement of input data by calculating the 1-qubit linear entropy of their amplitude encoding states. Then we implement an entropy matching approach to identify the optimal circuit depth that lead to the linear entropy being close the entropy of input data. The effectiveness of circuit designs based on entropy is verified by extensive experimental results. Specifically, we demonstrate that real-world datasets like MNIST images has limited quantum entanglement. Therefore, circuits designed with entropy matching exhibit relatively small depths being free from the barren plateau issue while maintaining benign performances in binary classification tasks. This work not only advances the efficiency of quantum circuit design but also sets the stage for further refinement of QML performance, with broad implications for practical quantum computing applications.

## 1 INTRODUCTION

Quantum computing is a new paradigm for computing based on quantum mechanics(Nielsen & Chuang, 2000), which has unimaginable advantages over its classical counterparts in many important applications(Shor, 1994; Grover, 1996). The physical implementation of quantum hardware has entered the Noisy Intermediate-Scale Quantum (NISQ) era, which is a significant milestone towards realizing practical quantum computing(Preskill, 2018). The current quantum computing devices, despite their noise and limited coherence times(), have demonstrated exponentially promotion in performing certain computing tasks over classical computers(Arute et al., 2019)(Zhong et al., 2020). Among the various approaches designed to exploit the power of NISQ devices, Variational Quantum Algorithms (VQAs)(Cerezo et al., 2021a) stand out as a particularly effective algorithmic framework. VQAs have demonstrated potential in a wide range of applications, particularly in machine learning, where they have been used to solve complex tasks with notable success.

VQAs are fundamentally composed of two parts: the classical optimization process that updates parameters and the variational quantum circuit (VQC) Peruzzo et al. (2014). The depth of the VQC is a critical factor in determining the algorithm's expressibility, which is its ability to model complex functions. Shallow circuits, while easier to train, often suffer from limited expressibility, restricting their capacity to solve more complex problems. Conversely, deeper circuits, although more expressive, can encounter the barren plateau problem, where the optimization landscape becomes flat, making training difficult. This creates a fundamental dilemma in VQC design: balancing expressibility with trainability.

In this work, we propose a novel approach to quantum circuit design that is tailored to the specific problem at hand. Our method involves using linear entropy as a metric to evaluate the entanglement properties of both the problem dataset and the quantum states generated by the VQC. By adjusting the circuit depth to match the linear entropy of the dataset, our approach strikes a balance between the expressibility and trainability of the quantum circuit. We validate our method through experiments on quantum binary classification tasks using multiple datasets, demonstrating that our entropy-matching approach leads to improved performance, effectively navigating the trade-offs inherent in VQC design.

## 2 PRELIMINARY

### 2.1 LITERATURE REVIEW

#### 2.1.1 QUANTUM CIRCUIT ARCHITECTURE DESIGN

Quantum circuit architecture design is a foundational aspect of developing efficient quantum algorithms and optimizing their implementation on quantum hardware(Farhi & Neven, 2018). The architecture directly influences a circuit's computational power, efficiency, and ability to generalize across various tasks(Preskill, 2018). Various architectural strategies, including shallow circuits, deep circuits, hybrid designs, and hardware-efficient ansätze (HEAs), have been explored to enhance expressibility and performance(Mitarai et al., 2018)(Kjaergaard et al., 2020)(McClean et al., 2016).

Shallow circuits, characterized by fewer gates and layers, tend to be more robust against noise and easier to train, making them suitable for near-term quantum devices(Cerezo et al., 2021b)(Bravyi et al., 2018). Conversely, deep circuits provide greater expressibility, allowing them to capture complex relationships in data. However, deep architectures often encounter challenges such as overfitting and the barren plateau problem, where gradients vanish during optimization, complicating the training process.

HEAs are specifically designed to utilize the capabilities of current quantum hardware effectively(Chen et al., 2021). These architectures typically consist of layers of single-qubit gates with tunable parameters and two-qubit gates to enable entanglement. By structuring circuits in this way, researchers aim to optimize computational resources while minimizing noise susceptibility. The trade-offs between expressibility and trainability are critical in designing circuits that can learn effectively from data.

#### 2.1.2 GENERALIZATION ERROR BOUND AND EXPRESSIBILITY ANALYSIS OF QUANTUM NEURAL NETWORKS

The performance of quantum neural networks (QNNs) depends significantly on their ability to generalize well to unseen data(Caro et al., 2022), a property that parallels the concept of generalization in classical machine learning. Generalization error bounds provide a theoretical framework for evaluating how well a QNN trained on a specific dataset will perform on new data. The generalization capability is influenced by several factors, including the architecture of the parameterized quantum circuit (PQC), the expressibility of the quantum circuit, and the complexity of the task(Cerezo et al., 2021a)(Abbas et al., 2021).

Expressibility can be quantified by comparing the distribution of states generated by a QNN with the uniform distribution of states, such as those from Haar random ensembles. A common method for quantifying expressibility involves calculating the Kullback-Leibler divergence between the fidelity probability distributions of the QNN and Haar random states(Sim et al., 2019). A lower divergence indicates higher expressibility, suggesting that the QNN can represent a wider range of functions effectively.

The interplay between expressibility and trainability is crucial in optimizing QNN architectures. Well-designed circuits should achieve high expressibility while maintaining effective training dynamics(Sim et al., 2019), allowing them to learn from the training dataset without succumbing to overfitting. Techniques derived from classical machine learning, such as regularization and model selection, have been adapted for quantum contexts to enhance the performance and generalization capabilities of QNNs(Cerezo et al., 2021a).(McClean et al., 2018)

## 2.2 QUANTUM COMPUTING KNOWLEDGE

Quantum computing(Nielsen & Chuang, 2000) is built upon fundamental concepts that bridge classical computation and quantum mechanics. At its core lies the qubit, the basic unit of quantum information. Unlike classical bits, which can exist in one of two states, 0 or 1, a qubit can exist in a superposition of both. Mathematically, a qubit's state is described as $|\psi\rangle = \alpha|0\rangle + \beta|1\rangle$, where $\alpha$ and $\beta$ are complex amplitudes such that $|\alpha|^2 + |\beta|^2 = 1$. This property allows quantum systems to process information in fundamentally different ways, leading to potential computational advantages.

Building upon this, multi-qubit systems are described using tensor products of individual qubit states. For an $N$-qubit system, the quantum state exists in a $2^N$-dimensional Hilbert space, and the overall state is represented by the vector $|\Psi\rangle = |\psi_1\rangle \otimes |\psi_2\rangle \otimes \cdots \otimes |\psi_N\rangle$. These quantum states are manipulated through quantum gates, the quantum analogue of classical logic gates. Quantum gates such as Pauli-X, Y, Z, and Hadamard, along with controlled gates like CNOT, enable the transformation of qubit states.

Quantum circuits provide the framework for organizing quantum gates to perform computations. A quantum circuit can be viewed as a sequence of gate operations applied to a set of qubits, transforming an initial quantum state into a final state. This process is parameterized in variational quantum circuits (VQCs), where the quantum gates depend on a set of classical parameters, denoted $\theta$. The task in many quantum algorithms is to optimize these parameters to minimize a cost function, typically defined as the expectation value of a quantum observable.

Measurement is a crucial aspect of quantum computation, where the outcome of a quantum process is obtained by collapsing the quantum state into a classical result. Measurements are represented by observables, which are Hermitian operators acting on the quantum state. For example, the Pauli-Z observable measures the probability of a qubit being in the state $|0\rangle$ or $|1\rangle$.

In quantum machine learning(Biamonte et al., 2017), loss functions quantify the difference between the predicted outcome and the actual target. These loss functions are typically defined in terms of the expectation value of an observable with respect to the quantum state produced by the quantum circuit. The general form of the loss function is given by:

$$\mathcal{L}(\theta) = \langle\psi_{\text{out}}(\theta)|\hat{O}|\psi_{\text{out}}(\theta)\rangle,$$

where $|\psi_{\text{out}}(\theta)\rangle$ is the output state of the quantum circuit with parameters $\theta$, and $\hat{O}$ is the observable.

To optimize the parameters of the quantum circuit, gradient-based optimization methods are often employed. In quantum computing, gradients can be computed using the parameter-shift rule. This rule provides an efficient way to calculate the derivative of the loss function with respect to the circuit parameters, without requiring explicit knowledge of the circuit's internal workings. For a parameter $\theta_j$ associated with a gate in the circuit, the gradient of the loss function with respect to $\theta_j$ is given by:

$$\frac{\partial\mathcal{L}}{\partial\theta_j} = \frac{\mathcal{L}(\theta_j + \frac{\pi}{2}) - \mathcal{L}(\theta_j - \frac{\pi}{2})}{2}.$$

This rule leverages the structure of quantum operations to efficiently compute the gradients necessary for optimizing quantum neural networks, enabling quantum algorithms to learn from data and refine their parameters over time.

# 3 THEORETICAL

## 3.1 HAAR MEASURE AND CIRCUIT EXPRESSIVITY

Haar measure is a type of invariant measure defined on a locally compact topological group $G$. The invariance property implies that for any element $g \in G$ and any measurable subset $A \subset G$, the Haar measure $\mu$ satisfies the following property:

$$\mu(gA) = \mu(A)$$

Here, $gA$ represents the set obtained by left-multiplying each element of $A$ by $g$. This means that the Haar measure is left-invariant on the group $G$, indicating that the measure of a set remains unchanged under translation by any group element. Similarly, Haar measure can also be right-invariant, satisfying $\mu(Ag) = \mu(A)$.

On a compact topological group (such as the unitary group $U(d)$), the Haar measure is unique (up to a multiplicative constant) and can be normalized as a probability measure, meaning that the measure of the entire group is 1. This normalization is particularly important because it allows us to define a uniformly distributed quantum state by randomly selecting a unitary matrix in quantum computing.

In quantum computing, we often focus on the unitary group $U(d)$, whose elements are unitary matrices in a $d$-dimensional Hilbert space. Unitary matrices are linear transformations that preserve inner products, satisfying $U^\dagger U = U U^\dagger = I$, where $U^\dagger$ is the conjugate transpose of $U$, and $I$ is the identity matrix.

The application of Haar measure on $U(d)$ is evident in the uniform selection of a unitary matrix $U$, which generates a quantum state $|\psi\rangle = U|\psi_0\rangle$ in the $d$-dimensional Hilbert space, where $|\psi_0\rangle$ is a fixed initial state in the Hilbert space. Due to the uniformity of the Haar measure, the generated quantum state $|\psi\rangle$ uniformly covers the entire Hilbert space, meaning that each possible quantum state is selected with equal probability, without any bias or inclination.

In practical computations, Haar measure is often expressed through integrals. For example, on the unitary group $U(d)$, Haar measure can be used to calculate the expected value of certain statistical quantities. Suppose we have a function $f(U)$ defined on the unitary group $U(d)$, then its integral over the group (according to Haar measure) is given by:

$$\int_{U(d)} f(U) d\mu(U)$$

For the purity of a quantum state $\text{Tr}(\rho^2)$, Haar measure can be used to calculate its expected value, where $\rho = |\psi\rangle\langle\psi|$ is the density matrix of the quantum state. In a $d$-dimensional Hilbert space, the expected purity of a pure state $|\psi\rangle$ chosen according to the Haar measure is:

$$\mathbb{E}[\text{Tr}(\rho^2)] = \frac{2}{d+1}$$

This result indicates that a quantum state randomly chosen from the Haar measure distribution has lower purity, meaning the state exhibits high randomness and complexity. This uniform distribution ensures that each quantum state is equally likely to be selected, demonstrating that the quantum circuit can generate a wide variety of possible quantum states, comprehensively and randomly covering the Hilbert space.

In quantum computing, the significance of Haar measure lies in its use as a standard for evaluating the expressiveness of quantum circuits. The expressiveness of a quantum circuit refers to its ability to generate a wide range of quantum states, which is crucial for performing diverse quantum computational tasks.

A key mathematical criterion for evaluating the expressiveness of a quantum circuit is the extent to which the quantum states it generates approach the distribution defined by the Haar measure. If a quantum circuit can produce quantum states that closely approximate the Haar measure distribution, it indicates that the circuit has sufficient complexity and flexibility to execute a wide range of quantum computing tasks, rather than being limited to specific states or operations.

Approximating the Haar measure distribution implies that the quantum circuit can realize arbitrary quantum states, which is essential for universal quantum computing. The Haar measure corresponds to a maximum entropy distribution, where the generated quantum circuit has the greatest information processing capability. Furthermore, if the quantum states generated by the circuit are close to the Haar measure, it signifies that the circuit can achieve highly complex quantum state transformations, demonstrating high expressiveness.

## 3.2 LINEAR ENTROPY AS A MEASURE OF QUANTUM CIRCUIT EXPRESSIVENESS

To quantify circuit expressivity, we can use a mathematical measure known as linear entropy, defined as:

$$S_L(\rho) = 1 - \text{Tr}(\rho^2)$$

where $\rho$ is the density matrix of the quantum state, and $\text{Tr}(\rho^2)$ represents the trace of the square of the density matrix.

Linear entropy is a measure of the degree of mixture or randomness in a quantum state. It provides insight into how "mixed" a quantum state is, with higher values of linear entropy indicating greater randomness and complexity in the quantum state. In the context of quantum circuits, a higher linear entropy suggests that the quantum states generated by the circuit are spread across a larger portion of the Hilbert space, closely approximating a uniform distribution defined by the Haar measure.

The Haar measure, in quantum computing, is a uniform distribution over the unitary group $U(d)$, where $d$ is the dimension of the Hilbert space. If a quantum circuit can generate states that are distributed according to the Haar measure, it implies that the circuit has a high degree of expressiveness and can explore a wide range of possible quantum states.

Linear entropy provides a straightforward and effective way to quantify the randomness and complexity of quantum states produced by a circuit. Specifically, it directly measures the purity or mixedness of a quantum state. A pure quantum state, which is fully coherent and exhibits no randomness, will have a linear entropy of zero:

$$\text{Tr}(\rho^2) = 1 \implies S_L = 0$$

In contrast, a maximally mixed state, where the quantum state is evenly distributed across all possible states, will have a lower trace and hence a higher linear entropy:

$$\text{Tr}(\rho^2) = \frac{1}{d} \implies S_L = 1 - \frac{1}{d}$$

Here, $d$ represents the dimension of the Hilbert space. The linear entropy thus ranges from 0 for pure states to $1 - \frac{1}{d}$ for maximally mixed states. This range provides a useful scale for assessing how well a quantum circuit can produce a distribution of quantum states that is close to the uniform distribution defined by the Haar measure.

When designing quantum circuits, especially for tasks in quantum machine learning, a key goal is to balance the circuit's expressiveness with its trainability. As the depth and complexity of a quantum circuit increase, so does its ability to generate complex quantum states. However, this increased complexity can also lead to what is known as the "barren plateau" problem, where the landscape of the optimization function becomes flat, making it difficult to find the optimal parameters using gradient-based methods.

The linear entropy serves as a diagnostic tool in this context. If a quantum circuit generates states with high linear entropy, it suggests that the circuit is highly expressive and can explore a large portion of the Hilbert space. This is beneficial for capturing complex data patterns but can also indicate a potential for optimization difficulties if the expressiveness is too high.

In practice, as the depth of a quantum circuit increases, the linear entropy of the states it generates tends to stabilize at a certain value. This stabilization often indicates that the quantum states have reached a level of distribution close to that defined by the Haar measure. From one perspective, this stabilization is a positive indicator, showing that the circuit is capable of exploring the high-dimensional quantum state space effectively.

However, from another perspective, it can be a warning sign. If the circuit's depth is too great, the resulting quantum states may be so randomized that the optimization landscape becomes too flat, leading to the barren plateau problem. In this scenario, while the circuit's expressiveness is high, its ability to be trained effectively diminishes.

Therefore, when using linear entropy as a measure of quantum circuit expressiveness, it is crucial to find a balance. The goal is to design a circuit that is expressive enough to capture the necessary complexity of the problem at hand, without introducing excessive randomness that could hinder optimization.

One practical application of linear entropy in quantum circuit design is determining the optimal circuit depth. By monitoring the linear entropy as the circuit depth increases, designers can identify the point at which the entropy stabilizes. This stabilization point often corresponds to the optimal depth, where the circuit is expressive enough to perform the task effectively without falling into the barren plateau trap. In binary classification tasks using QNNs, the process involves encoding all quantum states, calculating the linear entropy for single-qubit subsystems, and comparing the linear entropy for different circuit depths. By analyzing these values, one can determine the optimal depth at which the quantum circuit has sufficient expressiveness to distinguish between different classes of data while remaining trainable.

### 3.3 Determining the Optimal Circuit Depth Based on Entropy Matching

In the broader context of quantum computation, the quantum circuit $V(\theta)$ represents a dynamic and evolving framework where the interplay of parameters $\theta$ governs the transformation of an initial quantum state $|\psi_{\text{in}}\rangle$ into an output state $|\psi_{\text{out}}\rangle = V(\theta)|\psi_{\text{in}}\rangle$. This transformation encapsulates not just a technical procedure but the core essence of quantum machine learning, where the complexity of quantum operations mirrors the intricate nature of the information they process. The layers of parameterized gates, denoted as $L$, serve as fundamental building blocks, each subtly altering the trajectory of the quantum state through its vast state space.

As illustrated in Figure 1, the circuit often initializes all qubits in the standard state $|0\rangle$, but this seemingly trivial starting point belies the immense potential of the quantum states that follow. By encoding classical input data $x$ into quantum states through Y-rotation gates $R_y(\hat{x}_j) = \exp(-i\hat{x}_j Y/2)$, we introduce classical features into the quantum domain. However, this process transcends mere encoding—it is an act of transforming the very nature of data representation, where quantum states, unlike classical bits, embody a superposition of possibilities. This endows the quantum circuit with the power to represent and manipulate information in ways classical systems cannot. The structure of the quantum circuit, characterized by alternating layers of parameterized single-qubit gates and entangling operations (such as controlled-Z gates arranged in a circular topology), facilitates the entanglement and interaction of qubits, forming a web of quantum correlations. These correlations, unlike classical dependencies, are non-local, entangling the qubits in a manner that makes them inseparable in a quantum mechanical sense. This entanglement is the source of the quantum circuit's capacity for expressing complex transformations and interactions within the data.

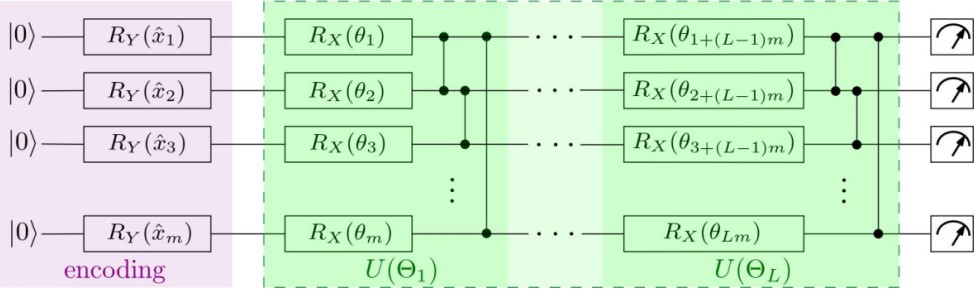

Figure 1: A typical example of a quantum circuit with $L$ layers of parameterized gates. All qubits are initialized in the state $|0\rangle$, followed by encoding the input $x$ using Y-rotation gates. The circuit consists of parameterized single-qubit gates and controlled-Z entangling gates applied in a ring topology.

When evaluating the circuit in the context of a binary classification task, we move beyond classical notions of data processing. Each input quantum state, representing the encoded dataset, undergoes a

transformation through the circuit. The resulting quantum state $|\psi_{\text{out}}\rangle$ is measured not only in terms of its final values but through the subtle lens of quantum entropy, specifically the linear entropy $S_L(\rho)$. This measurement provides a window into the degree of purity or randomness within the quantum system, with entropy reflecting the extent to which the quantum state remains coherent or has devolved into statistical noise.

In analyzing a dataset with different labels, we observe that the quantum circuit's ability to classify these states is tied to how the circuit balances complexity with stability. As the circuit depth $L$ increases, it gains the capacity to explore a broader spectrum of quantum states. However, this comes at a cost—excessive complexity can push the quantum system toward randomness, diluting the very structure it is meant to capture. This phenomenon is akin to the system's entropy approaching the Haar measure, a point where the states resemble a random ensemble.

The key challenge lies in navigating the trade-off between expressiveness and optimization. A quantum circuit that is too shallow lacks the capacity to explore the rich space of quantum states necessary for distinguishing between classes. Yet, if the circuit becomes too deep, the system may encounter the barren plateau problem, where the landscape of the optimization becomes flat and gradients vanish, stalling the learning process. The optimal depth $L$ is thus a delicate balance point, where the circuit is sufficiently complex to express the necessary transformations without succumbing to the randomness that hampers trainability.

By examining the entropy behavior of single-qubit subsystems across different circuit depths, we can identify this critical balance point. The entropy curves for states labeled +1 and -1 provide crucial insights into how the circuit processes and separates these states. When the difference in entropy between the two classes is minimized, the circuit is at its most effective, achieving maximum classification power without excessive entanglement or randomness.

Ultimately, this approach offers a pathway toward the optimization of quantum circuits, not just as a technical exercise but as a profound exploration of the boundaries between quantum randomness and structured information processing. It opens doors to more efficient quantum neural networks, where computational resources are judiciously conserved while achieving enhanced performance. The broader implications extend to the development of quantum technologies, where theoretical insights such as these pave the way for tangible advancements in quantum computing and its application to real-world problems.

# 4 EXPERIMENTAL RESULTS

## 4.1 THEORETICAL DETERMINATION OF OPTIMAL LAYER DEPTH

To investigate this, we conducted experiments using the MNIST dataset (Modified National Institute of Standards and Technology dataset). Each sample in the MNIST dataset is a grayscale image of 28x28 pixels. These images were first stored as 28x28 matrices and then flattened into vectors of length 784. Given that $2^9 < 784 < 2^{10}$, we designed a quantum circuit with 10 qubits, padding these vectors to 1024 dimensions and normalizing them to ensure that each vector had a norm of 1. The data was then encoded into corresponding quantum states $|\psi_{\text{in}}\rangle$ using amplitude encoding. For the experiment, 40 samples labeled as +1 and 40 samples labeled as -1 were selected, and the average linear entropy of the single-qubit subsystems was calculated for these samples. Preliminary results showed that the average linear entropy for samples labeled as -1 was 0.45316231711829386, while for samples labeled as +1, the average entropy was 0.4512322480823703.

The quantum circuit $V(\theta)$ used in this study, designed based on the theoretical model, is shown in Figure 2. For samples labeled as +1, the output state $|\psi\text{out}\rangle = V(\theta)|\psi\text{in}\rangle$ is expected to be close to the $|0\rangle$ state, while for samples labeled as -1, the output state is expected to be close to $|1\rangle$.

To precisely determine the optimal circuit depth, we computed the linear entropy of the single-qubit subsystems for the states $V(\theta)^\dagger|0\rangle$ and $V(\theta)^\dagger|1\rangle$ across different layers $L$. As the depth of the circuit increases, the distribution of quantum states tends to become more random, with the linear entropy approaching 0.5. This indicates that the quantum states have reached a distribution similar to that defined by the Haar measure. While this level of randomness suggests high expressiveness in the quantum circuit, it can also lead to the barren plateau problem, where optimization becomes difficult

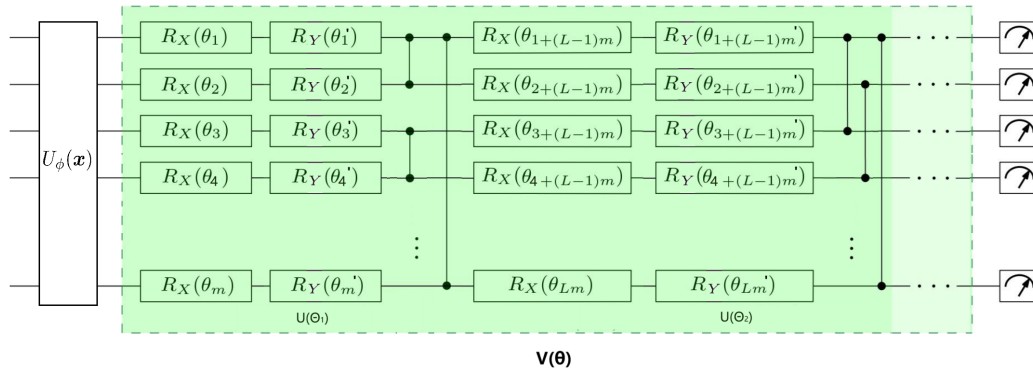

Figure 2: The quantum circuit used in the study. The circuit consists of $L$ layers, where each layer contains parameterized single-qubit gates and controlled-Z gates, arranged in a ring topology.

due to vanishing gradients. To refine the determination of the optimal layer count, we plotted the graph using $\ln(0.5 - \text{entropy})$ as the vertical axis, as shown in Figure 3.

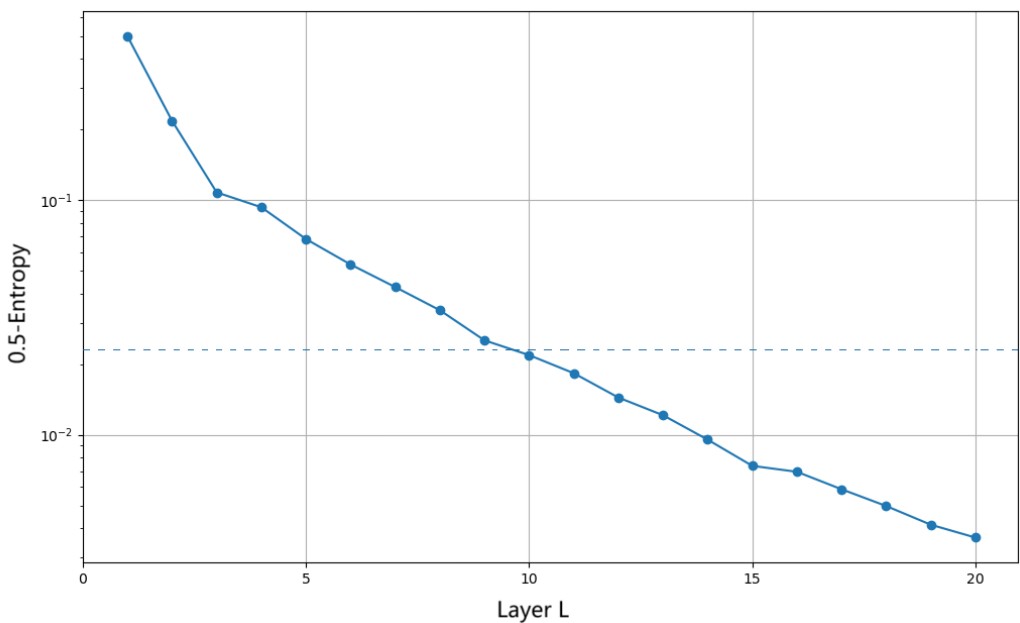

Figure 3: Linear entropy of the single-qubit subsystems across different layers $L$.

## 4.2 EXPERIMENTAL VERIFICATION

The previous analysis identified that the theoretically optimal circuit depth is 9 or 10 layers based on entropy matching. We proceeded with experiments to validate the accuracy of this theoretical prediction.

As shown in Figure 4, the experimental results demonstrate that by statistically analyzing the linear entropy of single-qubit subsystems of encoded quantum states and matching it with the linear entropy of quantum states generated by circuits of varying depths, the optimal number of layers for a quantum circuit can be effectively determined. In this experiment, quantum neural networks with circuit depths of L=9 and L=10 exhibited the best error rate convergence on the test set, validating the effectiveness of this method in optimizing circuit design. In contrast, while circuits with L=8

and L=11 also significantly reduced error rates, they fell slightly behind in terms of stability and convergence speed. Additionally, circuits with too few layers (such as L=1) showed slow error rate reduction and large fluctuations, indicating insufficient expressivity. Conversely, circuits with excessive layers (such as L=20) experienced slightly higher error rates, likely due to the barren plateau problem, which suggests that deeper circuits can lead to optimization difficulties and reduced computational efficiency.

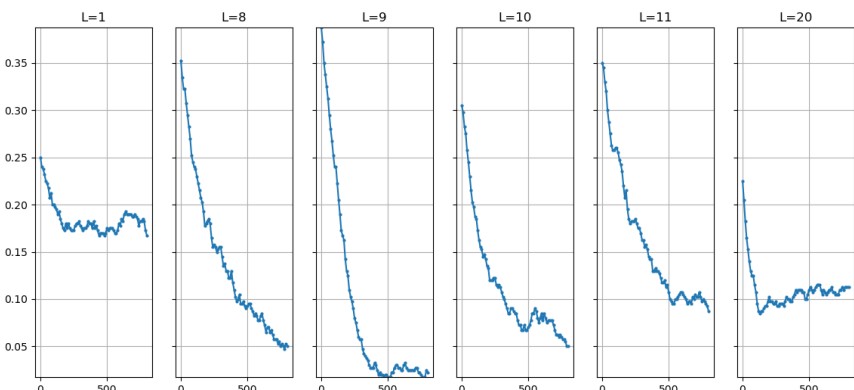

Figure 4: Error rates on the test set for circuits with different layers.

This method of determining the optimal circuit depth by matching linear entropy not only enhances the performance of quantum neural networks in classification tasks but also significantly conserves computational resources. By optimizing quantum circuit design, we can achieve superior performance in practical applications. These findings provide strong theoretical and experimental support for the future development of quantum computing technology, showing that designing and optimizing the structure of quantum neural networks by quantifying the complexity of quantum states is an effective approach to improving performance. This lays the groundwork for applying quantum machine learning to tasks such as classification and regression, further advancing the prospects of quantum computing technology in real-world applications.

## 5 CONCLUSION

In this work, we have explored the critical role of Haar measure and linear entropy in evaluating the expressiveness of quantum circuits, particularly in the context of quantum machine learning. The theoretical framework established in this study underscores the importance of ensuring that quantum circuits can generate quantum states that approximate the Haar measure distribution, which is indicative of a circuit's ability to explore the entire Hilbert space and perform complex quantum computations. Our approach utilized linear entropy as a diagnostic tool to quantify the randomness and complexity of quantum states produced by quantum circuits. By examining the linear entropy across various circuit depths, we were able to identify the optimal number of layers that balance expressiveness with trainability. This balance is crucial for avoiding the barren plateau problem, where excessive circuit depth leads to a flat optimization landscape, making it difficult to effectively train the quantum circuit. The experimental validation conducted on the MNIST dataset demonstrated that the theoretically determined optimal circuit depth, identified through entropy matching, significantly enhances the performance of quantum neural networks in binary classification tasks. The results showed that circuits with the optimal depth not only achieved lower error rates but also maintained stability across different test scenarios. The findings of this study have several important implications for the future of quantum computing. The entropy-based approach presented here offers a promising pathway for developing more efficient and effective quantum algorithms, with potential applications across a wide range of computational tasks. As quantum computing technology continues to evolve, the principles and methodologies discussed in this paper will play a pivotal role in shaping the design and optimization of future quantum circuits.

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
