# OpenReview forum: "Problem-dependent Quantum Circuit Design Based on Entropy Matching"
_ICLR.cc/2025/Conference — ICLR 2025 Conference Withdrawn Submission_

### Official Review · Reviewer_eZc6 · 2024-10-30

**Soundness:** 2
**Presentation:** 1
**Contribution:** 2
**Rating:** 3
**Confidence:** 3

**Summary:**

The paper considered the problem of architecture design in variational quantum algorithms, which is of importance in the area of quantum machine learning. The paper proposed using linear entropy for deciding the optimal circuit depth of the variational algorithms. Moreover, the authors conducted numerical experiments to show the effectiveness of their methodology.

**Strengths:**

The paper proposed a new way of using linear entropy for deciding the optimal circuit depth of the variational algorithms, with numerical experiments showing the advantage of their methodology.

**Weaknesses:**

- It seems that the authors uses the linear entropy in a more heuristic way with few theoretical justicfications. Could the authors explain more about why to use the linear entropy, not other measure like Renyi entropy? For instance, could the authors provide a a comparative analysis between linear entropy and Renyi entropy to show the advantages for variational quantum circuit design?
- In the experimental results section, more details could be provided. For example, could the authors explain more about how they compute the average linear entropy mentioned in line 367? A step-by-step description of how the average linear entropy was calculated, including any preprocessing of the MNIST data will be helpful for better understanding. Also, could the author provide more details about the experiments in section 4.2 about the training process? For example, it would be helpful if the authors could explain the optimization algorithm used, number of iterations, and how the error rates were calculated in these experiments.
- The writing and presentations in the submission could be improved. For example, in section 3.3 "DETERMINING THE OPTIMAL CIRCUIT DEPTH BASED ON ENTROPY MATCHING", it would be helpful to provide more diagrams to show the overall methodology and strategy for the entropy matching. Also, there are some typos, see the question part for more details.

**Questions:**

- Line 41, there should be no () after the word ''times''.
- Line 118, the $N$ qubit state is not always a tensor product state.
- Line 191, for pure state $\rho = \ket{\psi}\bra{\psi}$, the purity of $\rho$ should be $1$. Thus the equation for the expected purity of a random Haar state defined in the context seems to be wrong. Will this influence other claims in the paper?
- Line 371, the output state should be $\ket{\psi_{\text{out}}} = V(\theta) \ket{\psi_{\text{in}}}$.

---

### Official Review · Reviewer_ZWaF · 2024-11-03

**Soundness:** 2
**Presentation:** 2
**Contribution:** 2
**Rating:** 5
**Confidence:** 3

**Summary:**

This work proposes a novel method for determining the optimal circuit depth for variational quantum circuits (VQCs) based on entropy matching. Specifically, the authors match the linear entropy of the quantum circuit with the average linear entropy of the data samples to achieve a balance between expressivity and trainability. The linear entropy is calculated for different circuit depths to identify the point where the circuit effectively captures the relationships in the data before becoming too deep and encountering a barren plateau. The authors evaluate the proposed framework using MNIST images and demonstrate that the entropy matching approach successfully identifies the appropriate circuit depth.

**Strengths:**

This work addresses a crucial problem in variational quantum learning, determining the optimal circuit depth that balances performance and trainability on NISQ hardware. Using matching entropy to match the complexity of quantum circuits and data samples is a novel approach. The underlying theory of this method seems sound, and its evaluation using MNIST images supports its validity. The authors explain the concept of matching entropy and the necessary background knowledge in a clear and organized way. The matching entropy introduces a systematic way to guide quantum circuit designs based on specific problems/dataset. It also provides a way to mitigate barren plateu prior to training.

**Weaknesses:**

Linear entropy only considers one-qubit gates. To assess the power/complexity of a VQC, entanglement capabilities are essential as well. More metrics are needed to give a full picture of VQC's expressivity and computational power. For example, the Meyer-Wallach (MW) entanglement Q-measure [1] can be used to quantify the global entanglement measure of pure-state qubits entanglement.

Only one evaluation of a small subset of MNIST images are performed. More evaluations with a variety of dataset on real quantum hardware are needed to prove the robustness. MNIST images only consider binary image classification, which is a small subset of quantum machine learning applications. VQCs have other applications such as finding the ground state energy of molecules in quantum chemistry and portfolio optimization in finance. Testing the framework’s scalability and generalization to common quantum applications will further prove its feasibility. Including randomized benchmarking will also verify the validity.

Regarding the Haar measure introduction, more discussion of its usage/explanation in quantum computing field might be better than pure mathematical definitions. For example, explaining how single-qubit gates affect the Haar measure and including graphs could facilitate understanding.

[1] Gavin K. Brennen. 2003. An observable measure of entanglement for pure states ofmulti-qubitsystems. arXivpreprint,arXiv:quant-ph/0305094.

**Questions:**

1.	It is mentioned in the work that “the point at which the entropy stabilizes” is the point where the circuit has enough representation power without encountering barren plateu. Shallower circuits are less likely to fall into barren plateu, but how can we assure that the “stabilization point” of linear entropy of any quantum circuit can avoid barren plateu?
2.	In the MNIST experiment, the linear entropy of binary classification data is provided. How is it calculated, and how can we compute the linear entropy for other types of data, such as multi-class data?
3.	It is intuitive to think that when the linear entropy of a variational quantum circuit matches that of the data sample, the circuit has sufficient power to represent the patterns in the dataset. However, is there any mathematical evidence to support this correlation?
4.	In Figure 3, there is a blue dashed line that seems to indicate the chosen optimal circuit depth. However, the line is not labeled on the horizontal axis. What is the value of this line, and why was this specific value chosen as the optimal circuit depth?

---

### Official Review · Reviewer_PWex · 2024-11-03

**Soundness:** 2
**Presentation:** 1
**Contribution:** 1
**Rating:** 1
**Confidence:** 5

**Summary:**

The authors proposed an entropy-based method to evaluate the expressivity and trainability of parameterized quantum circuits. A more evenly distributed quantum state is believed to have smaller linear entropy, indicating a more expressive circuit and less trainability. Some numerical results are provided on the MNIST dataset.

**Strengths:**

NA

**Weaknesses:**

1. The manuscript provides limited information, as its 9 pages focus primarily on a single concept: linking the trace of the quantum state to the expressivity of the ansatz. This approach lacks depth and leaves much to be desired in terms of exploration of new ideas or techniques.

2. As far as I understand, the trace of a quantum state remains constant under unitary transformations unless the system interacts with an external environment. For any valid quantum state, the trace should remain 1, regardless of its distribution. Generally, the expressivity of a quantum ansatz relates to its ability to span the Hilbert space and requires the ansatz to remain unitary. It is unclear how the authors justify using entropy to assess expressivity, as entropy typically measures non-unitary characteristics. In particular, considering the quantum ansatz illustrated in Figure 2, it’s confusing where non-unitary transformations might be involved. Could the authors clarify this conceptual basis?

3. The experimental section lacks essential details. It is unclear whether the experiments were conducted on a quantum device or simulated on a classical computer. If simulations were performed, was circuit noise considered? Furthermore, the dotted line in Figure 3 is ambiguous; why does it indicate that layers 9 and 10 yield the best results? Additionally, the MNIST dataset includes images of digits 0–9, not +1 and -1. How was amplitude encoding performed as described, and what was the fidelity of state preparation?

4. The quantum circuit $V(\theta)$ should be a unitary matrix for a 10-qubit system. Therefore, it is unclear how $V(\theta)^\dagger$  could be applied directly to single-qubit states $|0\rangle$ and $|1\rangle$. This application requires further explanation to ensure consistency with the properties of unitary operators.

5. The manuscript’s technical writing lacks precision. Equations, numbers, and variables should be formatted in LaTeX math mode (e.g., "28x28" on line 362, $\psi$in and $\psi$out  on line 371, and instances of L in section 4.2). Additionally, line 344 mentions "The entropy curves for states labeled +1 and -1 ..." prematurely, as it relates to content in the following section. The figures lack proper captions and do not clearly label the x- and y-axes, making it difficult to interpret the results effectively.

**Questions:**

See weakness

---

### Official Review · Reviewer_cQF8 · 2024-11-04

**Soundness:** 1
**Presentation:** 2
**Contribution:** 1
**Rating:** 3
**Confidence:** 4

**Summary:**

This paper proposes a method to design quantum circuits which avoids trainability issues while retaining
sufficient expressibility to match the data. To this end, a design recipe is proposed to guide the construction of increasingly large
quantum circuits by measuring quantum entanglement, specifically in the form of entropy of input data with respect to single-qubit subsystems. This entropy matching creates a quantum circuit whose entropy properties are similar to the input data, thus, it is suggested, enhancing the learning. Numerical experiments show that the dataset MNIST exhibits low entropy.

**Strengths:**

The strong point of this work is in the aim to connect expressibility, which is already connected to trainability, to
an investigation for the entropy, which is an interesting idea. This results in more than plausible findings: it is not a good idea to provide too much depth, or in this case entropy, to the model. This proposal is mainly supported by numerical exploration.

**Weaknesses:**

1) I had a very hard time to understand the experimental results. For instance, the text explains that single-qubit entropies are
quantified (Fig. 3), but it is not specified for which qubit. If it is an average, there is no explicit mention to it, and no
error bars are provided nor is the overall behaviour explained.
2) It is not explained if such entropy is given for a particular configuration of the parameters or as a statistical measure - averaged over some quantities. This point is crucial in particular for this paper, because all barren plateau phenomena that the authors claim to study are precisely about the variances of the loss landscapes. The given presentation made it very difficult for me to properly understand the findings.
3) In general, the results could be made much more clear with proper explanations of the figures. For example, Figure 1
does not seem to correspond to the actual experiments. In Figure 3 there is a blue dashed line without any explanation. Figure
4 does not have axes labels, it is not clear what this plot represents.
4) Even on multiple reads, I did not fully understand the claims. It is mentioned that the average linear entropies of an amplitude-encoding data embedding is given by these numbers ∼ 0.45 (again with the same problems as before). Then, the entropies from data
and circuits are compared, and it is stated that those circuits with comparable entropies give optimal performances. Should it be clear why this is the case? Or is this what the experiments check? I do not see sufficient evidence to support these claims, a priori, but also when all the presented results are taken into account. Additionally, the text claims for theoretical optimization of number of layers, but I have not seen any such theory. In particular, in the section "EXPERIMENTAL RESULTS" the authors have the subsection "THEORETICAL DETERMINATION OF OPTIMAL LAYER DEPTH", which is a seeming contradiction (is it expermental or theoretical), and indeed it is not theoretical.

5) This paper assumes that expressibility is a guarantee for the existence, a priori, of a configuration of parameters with
good performance. While this is trivially true if expressivity = capacity to represent all states; but here expressivity is used as proxy for "emergence of barren plateaus" and this requires much less - just a 2 design, which do not cover a dense subset of unitaries... Thus claim is not supported by any known result in the literature, including the barren plateau saga. On the other hand, random states are known to exhibit high entropies and entanglements, so the results here stated are not surprising.

6) Topics related to this paper have been already explored, see for instance [1]. In this paper, the authors explore how
entanglement is built in a quantum neural network as the number of layers increases. This key reference is missing.

7) From a broader perspective, having to circuits with the same entropies for a given bipartition, does not guarantee
trainability or expressivity, since entropies are global properties of states. Much more information is required to recover
properties of the system.

8) From a presentation point of view, I find that there is too much dedication to concepts that are already well established,
such as entropy, quantum computing, parameter shift rule (whose presence I cannot connect to the content of the
paper), Haar-measure and so on. Also, some sections of the paper do not yield valuable information. For instance,
section 3 is called "theoretical ". Is there a word missing?

[1] Marco Ballarin, Stefano Mangini, Simone Montangero, Chiara Macchiavello, and Riccardo Mengoni. Entanglement
entropy production in Quantum Neural Networks

**Questions:**

Questions and suggestions –
• Is it possible to establish a strong connection between entropy and expressivity, as defined in the barren plateau
literature? Apart from the existing random states are generally entangled.

• The results would clearly benefit from a precise description of the obtained results.

---

### Note · Authors · 2024-11-14

I have read and agree with the venue's withdrawal policy on behalf of myself and my co-authors.